# Diet and Oral Health Coaching Methods and Models for the Independent Elderly

**Maria Antoniadou [1] and Theodoros Varzakas [2,\*]**

[1]  Dental School, National and Kapodistrian University of Athens, 15127 Athens, Greece; mantonia@dent.uoa.gr
[2]  Department Food Science and Technology, University of the Peloponnese, 24100 Kalamata, Greece
\*  Correspondence: theovarzakas@yahoo.gr; Tel.: +30-2721045279



**Featured Application:  Diet and oral health coaching is the emerging yardstick that will differentiate professionals, especially dentists, in order to become more effective as clinicians while it will guide the elderly patients to improve dietary habits, nutritional intake, and performance of oral hygiene for better oral health.**

**Abstract:**  Health-related behavior based on diet is an important determinant of oral health in independent elderly. Aging impairs senses, mastication, oral status, and function, causing nutritional needs and diet insufficiencies that contribute to a vicious circle of impairment. But the present needs of independent older adults suggest that health research and oral health care should shift from disease management and therapy to integral customized and personal treatment plans, including lifestyle, psychological, nutritional, and oral health coaching approaches. In this paper health coaching approaches in medical and dental settings are valued as to their effectiveness for older adults. Furthermore, coaching approaches for seniors are discussed and coaching models for better senior patient-dentist cooperation on the diet issue are suggested. Diet and oral health coaching is proven to be a modern senior patient-centered approach that needs to be incorporated at all relevant settings. It should aim to empower older adults in co-management of their oral diseases or bad diet habits affecting their oral health. This can be carried out through an incorporated educational plan for dentists either at the postgraduate or professional level since advantages seem to enhance the quality of life of the independent elderly.

**Keywords:**  diet; nutrition; oral health coaching; older adults; senior coaching; motivational interviewing; cognitive behavioral coaching techniques; independent elderly

## 1. Introduction

Aging impairs senses, mastication, oral status, and function, causing nutritional needs and diet insufficiencies. The present needs of independent older adults (OA) suggest that health research and oral health care should shift from reductionist disease management and therapy to integral customized and personal treatment plans, including lifestyle, psychological, nutritional, and oral health coaching approaches [1]. The American Society for Geriatric Dentistry, the Education Research Group of the International Association for Dental Research, and the American Association for Dental Research have been committed from their part to improving oral health in OA through education and skills development [2,3]. In response to these challenges, on the other part of the Atlantic, the European College of Gerodontology (ECG) and the European Geriatric Medicine Society (EUGMS) have created a common task and Finish Group. This group reported that the development of a workforce of dentists

with knowledge about and skills for working with OA would be enhanced by interdisciplinary and interprofessional education [4]. This philosophy has been also suggested by others in the past [5–7].

The vicious circle of malnutrition and oral health, discussed in detail elsewhere [1], should be broken especially for the independent OA, who may still be active or working. For those individuals who live alone or with family members, but still cooking and preparing meals by themselves, old recipes such as the Mediterranean diet (Med-Diet) should be kept as a base [1]. Then dentists or other medical professionals should enrich this base in a customized interpersonal way according to the specific needs of one's oral and general health status [1,4] and help people incorporate it in their daily routine. That is what oral health coaching does.

Generally, health coaching has been described as "the practice of health education and health promotion within a coaching context, underpinned by psychological principles, to enhance the well-being of individuals and to facilitate the achievement of their health-related goals" [8]. Health coaching is a patient-centered approach aiming, in other words, to empower patients in co-management of their disease or bad health habits (e.g., smoking, alcohol, diet, etc.) [9]. It is a strategy that emphasizes and supports patient autonomy, learning, and action instead of compliance. It is based on shared decision making and collaborative goal setting facilitated by motivational interviewing (MI) [8,10–13]. Basically, it is described throughout the literature as a partnership between the coach and the patient/coachee [13]. Cognitive behavioral coaching techniques and strategies are often used to tackle psychological blocks to goal achievement [14] by examining the patients' health-inhibiting thinking (HITs) and then helping them to develop health-enhancing thinking (HETs) [15]. But, as simple as that sounds, change is anything but easy for most of the people. It takes drive, motivation, action, and strategy to change one's habits—especially if someone has been doing things a certain way for a long period of time. For OA, the challenge is even greater due to physical and mental impairments that cause memory lapses or forgetfulness, lengthening of response, depression, loneliness, and even more anger and frustration because of aging. But as said by B. Pascal, "people are better persuaded to change by the reasons they themselves discovered than those that come into the minds of others".

Little data can be gathered on senior coaching concerning diet for oral health and those are mainly out of studies concerning interventions on nutritional aspects for diabetes mellitus, cardiovascular diseases, multi-morbidity, or cancer in independent adults, older adults, or frail elders in hospitals and care centers [16–21]. Thus, in this nonsystematic review the process of senior coaching on diet issues for better oral health are discussed for the independent OA. Suggestions, methods, and models for relevant senior coaching interventions are also described and compared. For purposes of this article, the term *older adults (OA)* refers to individuals age 65 or older.

## 2. Physical and Mental Issues of OA That Resist Change of Attitude

There are certain alterations in behavior during aging that may interfere in the process of desirable changes (Table 1). These alterations are: (1) Memory lapses or forgetfulness (symptoms might include varying degrees of memory loss, language difficulty, poor judgment, and communication, problems concentrating, and impaired visual perception). (2) Low mood after experiencing loss, coming with depression and a persistent feeling of sadness that can include changes in sleep, appetite, energy level, bad hygiene, and other areas. Mood changes, apathy, confusion, agitation, fear of death, or anger may also signal early dementia. (3) Discouragement or anger as health declines. Anger or aggression—which can show up as emotional or verbal abuse lashed out at loved ones or the doctors—can be particularly difficult to handle. (4) Takes longer to learn new things. On top of a normal decline in short-term memory in OA, it is also common to see a lengthening of "response time"—meaning they learn more slowly and retain new information less effectively. Many seniors who "age well" make a conscious effort to maintain mental alertness by reading widely, learning new skills, taking classes, and/or maintaining social contacts with people from a variety of age groups but this is quite often the exception for most of them. (5) No more resilience on life's hard modalities and various life events such as loneliness, death of a spouse, physical pain, lack of social life, estrangement

from family, eating by oneself, difficulty in getting foods, lack of cooking skills, and loss of economic independence [1].

**Table 1.** Physical and mental issues of Older adults (OA) that resist change of attitude.

| Physical and Mental Issues of OA | Symptoms | Results |
|---|---|---|
| Memory lapses or forgetfulness | Memory loss, language difficulty, poor judgment, absence of communication, problems concentrating impaired visual perception | Bad relationships Accidents Need of repetition Loss of orientation |
| Low mood or depression | Changes in sleep, appetite, energy level, denial and difficulty in oral and body hygiene | Unsocialized behavior, isolation, denial, estrangement from family |
| Sudden changes in mood | Apathy, confusion, agitation, fear, anger, breakdown | Difficulty or denial in supporting one's needs Social/role limiting |
| Discouragement or anger | Emotional or verbal abuse | Feeling of loneliness and fatality |
| Decline in short memory | Longer period of learning, Lengthening of response time Repetitive questioning | Loss of information, neglect of basic survival habits |
| Low resilience to pain and death | Lack of social life, estrangement from family, eating by oneself, loss of smiling and talking, loneliness | No visits to doctors & dentists, uncontrolled systematic diseases, high stress, bad oral health, anorexia |
| Lack of cooking skills and physical impairment | Difficulty in getting foods, eating only snacks | Malnutrition, bad oral hygiene |
| Loss of economic independence | Frustration, fear of the near future | Poverty, difficulty in getting foods, no access to health services |

## 3. Health and Oral Health Coaching Issues

Under the above discussed impaired circumstances, OA need customized repetitive and more motivational dietary interventions for general and oral health than younger individuals, in order to achieve desired changes. Most of all, it takes support, compassion, and empathy for facilitating any coaching approach in these individuals. Those are characteristics at which senior health coaching should excel to be effective.

As said before, the health coach-coachee/patient relationship is "a goal-oriented, client-centered partnership that is health-focused and occurs through a process of client enlightenment and empowerment" [11]. So, certified health coaches or health care professionals doing health coaching are somewhat like "change agents". They should understand how habits form, know how to reverse them, and specialize in helping people overcome obstacles in pursuing their goals. Their role thus involves listening, understanding, facilitating, applauding, supporting, motivating, providing feedback, rewarding, and helping the patient to weigh options and make choices. This can be accomplished by establishing trust and intimacy with the coachee/patient, active listening, powerful questioning, direct communication, creating awareness, designing action plans and goal setting with the coachee, and managing his/her progress and accountability [22]. In this process of change for the better, it is very important to identify and overcome challenges in the first place and then clarify the patient's strengths and aspirations, listening to his/her concerns, boosting his/her confidence in their ability to change, and eventually collaborating with him/her on a plan for change.

Health coaching, in specific, guides a learning process for improved disease or diet management that, if successful, it should lead to permanent changes in patient self-management skills and behavior. But these changes in self-management skills and behavior take time to influence health outcomes [6].

Therefore, in general, the impact of health coaching on health care and cost effectiveness should be assessed in long-term follow-ups [13] for all age groups but even more for the OA due to the physical and mental alterations discussed above.

The problem in the relevant literature is that evidence on the effectiveness of health coaching is, so far, conflicting and it is based on studies for adults with short-term follow-up only (up to 24 months) [6,13,23–26]. Due to the heterogeneity of target populations and outcome measures, no systematic reviews with meta-analyses have been completed [19]. So far, individual studies show basically either small or no significant effects of health coaching interventions [6,27]. They usually include the key recommendations shown in Table 2.

**Table 2.** Key recommendations for health coaching in OA.

| | Recommendations for OA during Health Coaching |
|---|---|
| 1 | Know how and when to call for help |
| 2 | Learn about the condition and set goals |
| 3 | Take medicines/nutrients correctly |
| 4 | Get recommended tests and services |
| 5 | Act to keep the condition well controlled |
| 6 | Make lifestyle changes and reduce risks |
| 7 | Build on strengths and overcome obstacles |
| 8 | Follow-up with specialists and appointments |

In many cases, self-management booklets are sent to patients to support progress toward the key recommendations [28]. Further, a traffic light system, telephone, or e-application can be used in order to visualize patients' progress and support [12,25–27,29–36].

Other research data reinforce the controversial benefits from diet health coaching in OA. To date, most of the large-scale lifestyle modification randomized controlled trials (RCTs) aiming to achieve healthy weight and/or improve nutrition were conducted among noncancer populations [23,37–39]. But, further, one should think that it is more interesting to evaluate the coaching effect especially on cancer patients. Since these patients are basically faced with the risk of death, they should be expected to be more willing to change habits. Generally, all cancer survivors are advised to adhere to the World Cancer Research Funds'/American Institute for Cancer Research (AICR) recommendations [28] to maintain a healthy weight, be physically active; eat a diet rich in fruits, vegetables, and whole grains; limit consumption of red and processed meats, sugar-sweetened beverages, fast foods high in fat, starches, or sugars, and alcohol; and do not rely on dietary supplements for cancer prevention. Additionally, it is recommended to abstain from smoking and reduce excess sun exposure. The American Cancer Society (ACS) guidelines for cancer survivors similarly aim to improve overall survival, metabolic health, and quality of life [40]. To one's great surprise, only a minority of cancer survivors meet the above ACS and AICR recommendations [41–46]. In a nationally representative survey among breast, prostate, and colorectal cancer survivors, only 16% to 18% consumed five or more servings per day of fruits and vegetables, and 24% to 43% engaged in 150 min or more per week of moderate to vigorous physical activity [44,47]. Also mentioned elsewhere, female breast cancer survivors are more likely than males to meet fruits and vegetables recommendations, while male cancer survivors are more likely than females to meet the physical recommendations [48]. Further, it seems that cancer survivors are more likely to adhere to recommendations either during cancer treatment or soon after completion of it [49]. A recent systematic review of lifestyle interventions among cancer survivors, including 51 studies, reported that cancer survivors' adherence to recommendations after participation in such studies is surprisingly low, at 23% on average (range, 7–40% [49]. The authors also reported that these interventions were more effective among survivors with diagnosis in the past five years or recent

survivors compared with long-term survivors (>5 years). Finally, survivors were more likely to adhere to recommendations to not smoke or to reduce alcohol consumption, while they were less likely to meet the recommendation for dietary fiber consumption, something that future senior coaches should keep in mind, too.

Reasons for cancer survivors not following diet and physical activity recommendations include lack of knowledge, low self-efficacy, and motivational and structural barriers (i.e., lack of access to healthy food and exercise facilities) to achieving sustained change [50]. On the other hand, a study showed that 80% of breast and prostate cancer survivors stated they are motivated to make lifestyle modifications through nutrition and physical activity health promotion programs [42]. So, data on this specific issue are quite controversial. It seems that, although patients are often provided enough, if not extensive, knowledge on diet and nutrition in order to change their dietary behaviors, they have only limited success in changing them [50]. It is important to mention that, although initial changes may occur, these may not persist over the long term [51,52]. Everywhere in the literature it is highlighted that patient self-management is not always easy to accomplish. It is difficult to change a long-entrenched lifestyle, even when there is motivation to do so; however, it is much more difficult if there is no motivation. Psychosocial and financial factors are key barriers especially for OA. Many of them, usually quite independent during their lifespan, may be embarrassed about the need for help, lack resources to make changes, or may fear failure and the associated perception that they are incompetent. Of course, there has often not been a strong support system within the medical community to help OA to manage on their own nor in the community at large or even sometimes within the family. To address this gap, effective lifestyle modification programs at the clinics, dental units, and community centers and settings are needed to promote sustained behavior change for those individuals [24,27,53].

It is thus important to conclude that, so far, adherence of this aging group to professional recommendations is astonishingly low. Of course, there always seems to be a gap between what people 'know' and what they 'do'. The process that maintains the gap between knowledge and behavior is ambivalence. OA are faced with conflicting motivations and pressures; the change feels too big, the rewards too distant, motives no longer exists, the personal or financial costs are too high, or maybe it was never their idea to change in the first place [18]. Studies on adherence to health professionals' recommendations have shown that approximately 30–60% of health information provided in the clinician–patient encounter is forgotten within an hour and that 50% of health recommendations are not followed [54]. Thus, overcoming persistent noncompliance of OA can make health-behavior change one of the most rewarding and the most challenging responsibilities for dental health professionals.

## 4. Positive Data on Health Coaching

Health coaching has led to positive patient outcomes in several studies, including weight loss, diabetes control, decreased blood pressure, HIV, and improved health behavior and multi-morbidity [25,30,55–61]. Previous research has also demonstrated that when used in chronic disease management, wellness coaching enhances self-management skills in patients with diabetes and helps reduce readmissions in those with chronic obstructive lung disease [62,63]. In a review of 15 randomized health coaching interventions, six were able to demonstrate significant improvements in targeted behaviors such as physical activity and medication adherence [64]. It was also reported that wellness coaching was associated with improvement in three areas of psychosocial functioning: Quality of life, mood, and perceived stress [65]. Participants also improved their self-reported health behaviors and goal-setting skills [66]. It was further suggested that integrating wellness coaching within primary care practice is a feasible model for diabetes care, which can be done even without significant additional resources [67]. It is also reported as being well received by patients and physicians in primary care setting [68]. While wellness coaching conducted in health care settings has been shown to be effective in chronic disease and weight management [35,62,63], its use among OA who do not have a chronic disease but who are at high risk for it has not been widely explored. In the study of Knowler et al. [16],

the methodology focused on a well-structured curriculum that included supervised physical activity sessions supported by individual case managers who functioned as "lifestyle coaches". Also, in a systematic review of counseling interventions to change diet and physical activity behaviors among obese and overweight persons with cardiovascular disease risk factors, it was reported that there was decreased diabetes incidence and improved intermediate cardiovascular health outcomes up to two years [57].

On the part of nutrition and diet, clinical recommendations guide clinicians to support especially cancer patients in making healthy diet and nutrition choices [69,70]. Clinical assessments can provide a snapshot of the current state of dietary consumption and dietary patterns of those patients. Most dietary interventions in cancer patient populations exist within a framework of lifestyle interventions for diet and physical activity as well as weight loss [71]. Specifically, targeted dietary change interventions focus on either weight loss or encouraging weight maintenance in cancer patients with good results [71].

Prior dietary interventions have included and tested behavioral models to improve understanding of how patients change their behavior. Successful interventions have used behavior change techniques derived from theoretical behavior models [72,73]. Further, dietary change strategies have been identified to manage weight in cancer populations [74,75]. Two recent reviews [71,74] demonstrated the relevance of the social cognitive theory (SCT) behavioral model [76]. In the study of Park and Chang [61], the effectiveness of a health-coaching self-management program for OA with multi-morbidity in nursing homes was studied with success. Participants in the intervention group had significantly better outcomes in exercise behaviors, cognitive symptom management, mental stress management/relaxation, self-rated health, reduced illness intrusiveness, depression, and social/role activities' limitations. In addition, there was a significant time-by-group interaction in self-efficacy. According to the goal attainment scales, their individual goals of oral health and stress reduction were achieved.

It is also reported elsewhere that health coaching has the potential to decrease the amount of time patients spend with a physician, decrease physician follow-up, and increase satisfaction among both patients and providers as a result of the delivery of more personalized care [77,78]. As an example, one study reported increased patient trust in their physician when health coaching was provided [79].

Furthermore, evidence demonstrates that targeted motivational interviewing in the treatment of chronic diseases and conditions prevalent in OA achieves positive outcomes and reduces health-related costs [80]. It was also reported that when patients receive collaborative self-management support, they have fewer hospitalizations, improved quality of life, and improved clinical outcomes in several ambulatory-sensitive conditions [81–83]. Especially, health coaching provided by nurses has shown promise as a strategy for facilitating behavior change that can lead to improvement in OA with chronic illnesses [84]. It is reported that based on a humanistic and holistic perspective, health coaching is compatible with nursing ideals, and a coaching strategy holds promise for helping OA to achieve their health goals [9]. Coaching by nurses may motivate OA with chronic illnesses to move forward, to act towards making lifestyle changes, and to increase their understanding [85]. So it seems that health coaching could be an expected competency not only for nurses but also for dentists, doctors, and other medical professionals who could help OA promote their self-management skills, prevent complications, lessen their health and oral health care costs, and appreciate a better quality of life [9,86,87].

It is further reported elsewhere that OA are more likely to benefit from a series of health education sessions followed by tailored feedback from the counselor [88]. All that is needed is absence of criticism, patience, empathy, and total acceptance by the dentist/professional coach. Empathy is said to be the necessary element for effective communication between patients and providers to achieve optimal clinical outcomes. Empathy has been defined as a "predominantly cognitive attribute that involves an understanding of patients' experiences, concerns and perspectives combined with a capacity to communicate this understanding and an intention to help" [89–91]. Higher empathy scores have been positively associated with clinical competence and better patient outcomes in physicians [92]. The nature of empathy has been studied extensively in medical students but less so

in dental students [93]. Sherman and Cramer (2005) [94] found that the psychometric properties of empathy in a sample of dental students were comparable to those found in medical students [94]. Four factors emerged, such as perspective taking, compassionate care, standing in the patients' shoes, and efforts to ignore emotions in patient care. Waldrop et al. [7] studied dental students' knowledge about aging and found that, although information is readily consumed by dental students, positive attitudes are not as easily taught [95,96]. It was also reported that attitudes are significantly influenced by the amount of exposure to older people [97], but that is not the case elsewhere [7]. However, attitudes and knowledge may only partially contribute to the development of a caring professional [7]. It is interesting to know that women are scoring higher in empathy than males among dental students concerning communication with OA [7,94].

After all the above, it seems that dental and medical professionals should spare time to explore the factors mentioned before and attach them to the character of the OA. Then oral health coaching based on empathy could be very effective in encouraging, inspiring, and empowering them to reach their maximum health potential [98].

To do so, professionals need training in coaching strategies [9,99,100]. For this reason, coaching modalities were sparingly investigated for their effectiveness in the program of studies both in dental and medical schools with promising effects [101–103]. In a study where medical students were enrolled in the role of health coach for patients with diabetes it was shown that patients accepted the procedure as an opportunity to learn a great deal about the management of their diabetes. Several participants mentioned that the student was so persistent that they eventually altered their exercise and dietary behaviors. In addition, several patients mentioned they often felt uncomfortable asking their regular physician questions due to time constraints. Because of their relationship with student health coaches, patients expressed feeling more comfortable talking to their coach who, in turn, would obtain answers to their health questions. In addition, several patients mentioned that working with students on their health goals improved their motivation to change health behavior [104]. However, although the use of medical students as health coaches to increase patient activation is a novel approach, there seems to be an indication that health coaching by medical students can improve health care communication and disease awareness among patients regarding their disease and overall health [104]. No such data exist yet on dental education, making it a promising research field.

Generally, it can be assumed that improvement in communication between patients and their health care providers can allow for higher utilization of health care, better adherence to treatment recommendations, and improved management of chronic disease, such as diabetes [53,104–107]. From the above, the suggestion was derived that dental and medical professionals could be trained to serve as health coaches [108] with great success if time is found for them to be educated on and perform it, together with their other responsibilities.

## 5. Methodology for Behavior Change during Coaching

Although no single theory or conceptual model dominates health behavior research or practice of coaching [109], it is well recognized that interventions to modify health behaviors are enhanced through reliance on health behavior theory [72,87,110–112], including foundational behavior change theories such as social cognitive theory [113], the health belief model [113,114], the theory of reasoned action and the theory of planned behavior [115], the integrated behavioral model [116], the precaution adoption process model [117], health locus of control theory [118], and the transtheoretical model of behavior change [119]. Due to overlap among these and other foundational theories, and because only a limited number of variables are relevant to consider when promoting health behavior change [120], Fishbein proposed the integrative model [55] to unite a volume of theory from years of interdisciplinary work into a coherent model to support health behavior change practices [120].

Further to be discussed here, social cognitive theory is a unified conceptual framework, which taps into patients' beliefs in their capability to engage in a new behavior and their expectations of how engaging in that new behavior will influence their health (i.e., the outcome of interest) [121]. Beliefs about capabilities, beliefs about consequences, and social influence are important determinants of adopting and maintaining dietary behavior change. These beliefs and attitudes are then targeted by behavior modification techniques (i.e., the intervention), which then leads to changes in behavior and subsequent changes in health outcomes [76].

The Transtheoretical Model stages of change construct complements by describing the five stages individuals move through as they make behavioral changes [122–125]. It has been effectively used to target and adapt behavioral interventions and to measure the magnitude of effective interventions. This theory has been used extensively across cancer survivor populations, within different cultural settings and applied to variety of behaviors (e.g., diet, physical activity, and weight management) [71,74].

Approaches to behavior change broadly consist of individual- and group-level interventions, and a combination of approaches has been shown to be more effective than one approach or the other [74,75]. Changing dietary behaviors in cancer patient populations adopts variations in behavior change models, with success being driven by a unique combination of behavior change techniques. Five general techniques frequently emerge as effective within published interventions: Goal setting, action planning, social support, instruction on how to perform behavior, and motivation. Self-monitoring of behavior and feedback on behavior are common in interventions, but these techniques were less effective [75].

Behavior change models attempt to explain why patients may change their behavior, with an emphasis on how these internal and external factors mediate the relationship of change to improve health outcomes (e.g., diet). However, most successful interventions for cancer patients include not only dietary change but also physical activity and behavior modification support in the form of materials to assist in change [126]. Lifestyle behavior modification interventions have previously focused on cancer survivors, but more recently a change has taken place to shift focus to supporting patients with dietary change during treatment.

Therefore, targeting theory-based factors is improving dietary and physical activity lifestyle interventions in cancer patients, although additional development is necessary to inform better intervention programs for longer-term maintenance of weight change. Evidence exists on the benefits of such interventions to achieve and maintain healthy weights and to adhere to nutrition and physical activity recommendations for improving cancer prognosis and survival. Examples included below demonstrate first the need to make sure there are multiple strategies to support behavior change, as patients have differing needs. Second, in cancer survivors, targeting behavioral motivation factors (i.e., self-confidence, goal setting, self-monitoring, feedback, taste preferences) can improve healthier food choices. Third, there is a need to consider more pragmatic approaches using adaptive communication strategies in person and via electronic messaging (i.e., text messaging, interactive websites). In the study of DeJesus et al. [34] the coaching methodology that was followed was 12 weeks of one-to-one coaching conducted mainly on a face-to-face basis. Alternative methods of delivering behavioral interventions by web or mobile devices are showing promise [25], as well as a combination of personal and group coaching [30]. Other wellness coaching studies mention that the duration of the intervention program [16,17,122] consisted of at least 12 weeks of sessions, while even shorter duration, such as six weeks or shorter, posed also feasibility with significant changes in outcome measures [53].

Furthermore, the implementation of the new technology seems promising in achieving this. For example, Kima et al. [53] developed a new mHealth version of "the Self-Help Intervention Program (SHIP)," by incorporating the principles of persuasive technology [29]. In addition, to address the relatively slow "technology readiness" of the target population, they incorporated human interaction into the intervention using community health workers (CHWs) as facilitators. This hybrid intervention, called model hSHIP, which combines digital and human touch, was inspired and influenced by the collective work of B. J. Fogg, who coined the term "persuasive computing "(later broadened to "persuasive technology"), and his colleagues at the Stanford Persuasive Technology Laboratory.

Persuasive technology is a new, evolving branch of implementation science that acknowledges the ubiquitous yet invisible influences of technology on behavioral change. Fogg postulates seven primary task support principles that, when incorporated into systems, applications, and technologies, support and enable behavior change without coercion [99]. For the hSHIP, a chronic disease management system (CDMS) was developed that combined all processes of project management (recruitment and enrollment, monitoring, questionnaires, messaging, reporting, etc.) in real time and delivers the intervention's components (education and training, monitoring and counseling, messaging, goal setting, etc.) into a web application. Thus, research nurses and CHWs communicated with program participants in real time using smartphone modules for Short Message Service (SMS) and notifications in the CDMS. The findings suggested that it is possible to sustain motivation to engage in self-care behaviors over the long term, so that those behaviors will be translated into optimal clinical outcomes. The key to sustaining motivation is constant and immediate feedback through a combination of digital and personal touch, because positive, real-time feedback helps to eliminate uncertainties, fear, or reluctance in self-care behaviors. Furthermore, utilizing the most innovative technology in an accessible, personalized, self-help intervention that will proactively reduce potential heath disparity gaps is consistent with the movement towards precision medicine/health [53].

So far it seems that continued efforts to further refine wellness coaching programs through new technological interventions will help optimize their role in OA health prevention measures [72]. Also, it is unlikely that there is any or only one health theory that works ideally to promote health in all contexts, by all providers, for all types of OA [127]; further, all theories are not constant but in flux and evolving over time [128].

## 6. Oral Health Coaching

The benefits of oral health coaching, however, have been reported mostly anecdotally. Complete understanding of effective behavior changes in the dental setting and coaching research, especially in OA group, is in its infancy [72,87,112,122,127,129]. Moving forward, it is important to learn what behavior change approaches work best in the dental setting, as well as for whom, how, and when such approaches work [128,130]. This will require study designs that can measure, isolate, and validate health theory mechanisms of action [127]. To date, it is understood that, in the dental and other health care settings, providing information alone appears to have little long-term impact on promoting behavior change [122,131]. Why is this? It is because this kind of approach is based on many assumptions—e.g., that people want to know this information (they perceive it as being relevant and important to their lives); that they understand this information; that they are ready, able, and motivated to apply this information; and, further, that they can address any challenges that should arise in implementing this information both in the short and long term.

The contemporary field of oral health behavior and oral health education has shifted considerably and now reflects a blending of theory, strategies, models, and approaches between the social, dental, and medical sciences. Additionally, theory and research on integrative health coaching and intentional change coaching suggest that it is critical for the provider to communicate hope, trust, and genuine optimism to the patient (both verbally and nonverbally) in order to ground the provider-patient exchange in the patient's intrinsic hope, motivation, and vision of health and well-being [100,132]. Even in an emergency care condition, the provider can "plant the seeds" to raise the patient's oral health self-awareness in the near- or long-term future. Indeed, establishing a connection based on shared hope, trust, and respect may enhance the likelihood that the patient will return for follow-up (and, ideally, ongoing routine) dental care [98,133,134]. This increased interest in the psychosocial aspects of behavior change was evident in a recent systematic review of interventions to improve oral hygiene based on psychological models [135].

*6.1. Oral Health Coaching Techniques and Models for OA*

The general layout needed for an effective oral health senior coaching intervention, addressing diet needs for better oral health, should be based on certain characteristics employed by certified coaching associations, like International Coach Federation (ICF) [22] or Association for Coaching [136], translated, and incorporated into the oral health sector from dental professionals.

Then the basic procedure for engaging patients in self-management for better diet habits towards better oral health should be: (1) *Preparation for the visit*. It is important for both the patient and the dentist to prepare for the visit. Patients who can share their concerns with a care coordinator or provider are less anxious and show more improvement, even if they just provide a written list of those concerns. So, it is crucial for the dental professionals to help patients understand their central role in managing their conditions and that the entire health care team is there to help. Time must be found for the practice of self-management by gathering clinical and patient experience data in the chart and encouraging patients to bring questions and concerns on their next visit. (2) *Scheduling an agenda together*. At the start of the visit with the patient, a list should be written down of the things that each of the parties hope to achieve during the visit and prioritize the most important items needed to be addressed first. Working together to build the agenda demonstrates that the patient's concerns are valued, and time will be given in order to hear them. The patient feels appreciated, which is a strong motivational feeling. (3) *Asking open-ended questions.* Encouraging the patient to share their experience with the dentist by asking questions that require more than a yes/no response. This can also be done in the form of a statement such as "Tell me more about that". (It will be discussed further later.) (4) *Practicing reflective listening in order to build a trusting relationship with the patient*. It is important to practice reflective listening, without interruption, and respond by rephrasing what it was heard without adding meaning or judgment. (5) *Recognizing and eliciting "change talk".* Change talk is any statement that expresses a desire to change. The professional "catches the moment" in order to enhance possible alterations in behavior. (6) *Affirmation and celebration of what works.* At the end of the session and during the recall appointment of the patient, the dentist should make time to acknowledge and talk about what has worked and what success will look like. Time should be spent in discussing how it will look and feel to accomplish the patient's goals. Celebrations of even small achievements are crucial in the coaching process, making people feel proud of themselves and enhance their motivation. (7) *Making a specific and realistic plan.* Identification of the concrete steps that will be taken to address diet and oral hygiene habits should be made in partnership with the patient. Discussion of the different options and selection of the best one that is consistent with the patient's lifestyle and that the patient is confident he/she can implement should take place. Also, there should be a timeline and talking about the way that the monitoring of the progress will take place. This procedure builds patient's confidence in his/her ability to reach these goals. A written care plan or visit summary, which includes goals and action plans and ensures patients and families on what to do when they leave the visit, should be made. (8) *Following up.* There should always be time for arranging support services that will help the patient to be successful in achieving his/her goals. For some people this may mean a phone call in the next 24–48 h, while for others a follow-up phone-call in 1–2 weeks [122,136–139].

6.1.1. Motivational Interviewing in the Service of Senior Oral Health Coaching

The use of motivational interviewing (MI) is suggested in health settings [1,122]. MI is a person-centered, goal-directed method of communication for eliciting and strengthening intrinsic motivation for positive change [140]. It is predicated on a 'spirit' of rapport, based on partnership, empathy, and acceptance. As such, the MI counselor must be willing to hear, accept, and respond to a patient's personal perspective rather than recite a predetermined set of prescribed instructions and guidelines. It is important to mention that the information-giving approach seems to have no effect on behavior change but behavior change with self-monitoring and goal setting is a better approach [135]. A key component of a MI conversation for OA is to acknowledge that they have every right to make no change. Acceptance of the situation as it is does not mean though that a guiding communication style,

which invites people to consider their own situation and find their own solutions to situations that they identify as problematic, would not work towards change [141]. The patient's view is elicited by the clinician in order to help them understand the situation from the client's perspective including their goals and values. This is a collaborative approach in which the expertise of the practitioner plays a part, but it is the patient's journey as he/she decides where to go and if and how to get there [18]. MI has shown good results in different dental settings [135,142]. However, these results are transitory, have negligible impact on the incidence of dental caries [133,142], and are not yet searched for effectiveness in OA.

OARS Model in MI

A MI model well discussed in the relevant literature, supposed to work in OA, is OARS (Open-ended Questions-Affirming-Reflective Listening-Summarizing). OARS is the acronym for the four core communication skills – open-ended questions, affirmations, reflections, and summary—that are integral to the collaborative, client-centered, motivational interviewing approach. While many of these are not new concepts, their collective and strategic use is the essence of the spirit of MI. As described above, many aspects of the dental visits are routinely closed-ended. Medical history questions seek yes/no answers, whereas the types of oral hygiene used generate short categorical responses. Often in response to the presence of disease, traditional instructions consist of a prescribed explanation of the disease process that is entirely a one-way communication. By comparison, the OARS approach not only provides key strategies to shift the conversation so that the patient is doing more of the talking, it also provides an opportunity to discover what is uniquely meaningful to each individual, gauging their oral health understanding and their desire and ability to change. This is important for OA who seek acceptance and trust in order to overcome their fears. To achieve this insight, it is necessary to use the OARS approach to get OA talking. The R in OARS stands for reflective listening. Reflections of patients' responses to open-ended questions serve two main purposes. First, it develops the partnership by showing the patient we really hear what they have to say. The intent is to listen for responses that represent change talk (in the direction of the desired change) or sustain talk (avoiding change) that will be discussed further in other models. Second, if the provider is unsure, he/she understood the patient correctly, it provides an opportunity to clarify meaning. Varying levels of reflections, from simple repetition of what the client said to amplified reflections that exaggerate the response, help direct the patient in the direction of the health-behavior change wished to achieve. Skilled reflections allow the provider to interpret the meaning of the patient's responses [141]. It is important, though, that the reflection is made as an interpretive statement, not a question. A good method to use when beginning to use reflective statements is the phrase, "*Sounds like . . .*" (e.g., "Sounds like the deep pockets worry you"). Once the provider become accustomed to using reflections, he/she can simply drop the 'Sounds like' expression.

The S in OARS is summary. Summaries reiterate the fact that the dentist was truly listening, while setting the stage for behavior change. The art is to summarize any aspects of the conversation, allowing OA to hear any contradictions in their own responses with a focus on what they want to do next.

The four processes of OARS model—engaging, focusing, evoking, and planning—strengthen a patient's own motivation for, and commitment to, change. Beyond OARS, more sophisticated motivational interviewing strategies are aimed specifically at evoking and planning intrinsic motivation for behavior change. Once again, the plan ultimately originates from the patient with direction from the clinician [142]. Decisional balance is another useful strategy for evoking and planning. It is a means of allowing the patient to examine the pros and cons of a behavior change. This strategy is particularly helpful for OA who are ambivalent, uncertain, fearful, or reluctant about making a change [110,142].

6.1.2. Other Models and Tools for Immediate Senior Oral Health Coaching

For time-management reasons, in a private dental practice or senior center, dentists and other professionals acting like coaches should be introduced only to certain models for quick patient interference, like the following.

Dental PAM (Patient Activation Measure)

Patient activation (PA) refers to a person's ability to manage their health and health care. An activated patient has knowledge, skill, and confidence to manage his/her health and health care in wellness and illness. The chronic care model was built with the understanding that the patients would learn how to manage their care on a day-by-day basis. However, the level of PA varies considerably, as was mentioned already. Clinicians, so far, strongly encourage patients to follow medical advice but are less likely to endorse that patients should be able to make independent judgements or take independent actions. Discussing a practice's culture around patient self-management is a critical first step. Stratifying patients according to activation level using the evidence-based tool, Patient Activation Measure® (PAM®), provides an effective method to: (1) Guide resource allocation at the practice level, (2) tailor support to a patient's abilities, and (3) improve patient safety and satisfaction [143]. The Patient Activation Measure (PAM) is a global assessment of an individual's self-management competency. PAM quickly evaluates three key personal health domains—knowledge, skills, and confidence—and segments patients into one of four activation levels along an empirically derived continuum. Coaching for activation focuses on seven core areas of self-management—condition and symptom understanding, medication adherence, diet and nutrition, physical activity, stress and coping, information seeking, and smoking cessation. Each area of self-management is tailored to health status, addressing diabetes, asthma, Chronic Obstructive Pulmonary Disease (COPD), Congestive Heart failure (CHF), Coronary Artery Disease (CAD), hypertension, and high cholesterol, as well as disease prevention through a lifestyle module. Within each self-management core category, information, goals, and related action steps are tailored to an individual's health status and level of activation. Goals and steps are supported with self-care resources suitable for coaching use. Potential contribution of an oral health PAM instrument–not yet implemented-is suggested for the dentistry field [143].

Tell-Show-Do

The most common model in dentistry, well proposed in children and young adolescents, is the model of tell-show-do. The steps described for this model are: (1) Tell or explain the procedure, (2) show or demonstrate the procedure, and finally, (3) the learner can do or practice the technique until he/she has mastered the skills involved. The last step is the most important one if the learner is to develop proficiency. For OA, this model is expected to bring direct conscientiousness of the present situation and is simple to practice for both the dentists and the patients [144]. It will be well performed in cases of memory weakness and in OA with sudden mood changes or depression.

Balloons' Diagram

In the balloons' diagram [145], the patient can place in the balloons the problems and worries he/she must face in order to release them one by one. Some helpful questions might be: *"What do you think is going on?"* or *"What is your understanding of this (condition, issue)?"* or *"What worries you the most?"* or *"What else are you concerned about?"* or *"What do you know about (treatment, self-management)?"* or *"Which balloon do you want to release first?"* The balloons' form could be provided to the patient at the reception area; so the front office staff should be trained in introducing the form and asking questions like: *"Which of the healthy change activities seems most important to you right now?* or *"Which bad diet habit could you put into the balloon to release first?"* (if none does, ask what other area they might choose to address) or *"We are working on improving our care for people with (mention the oral health problem). Dr. X would like to discuss your health goals with you. This form has some ideas you might consider placing in*

*these balloons.*" Since this model engages the vision, it might be effective for OA with other physical impairments such as listening or memory problems (Figure 1).

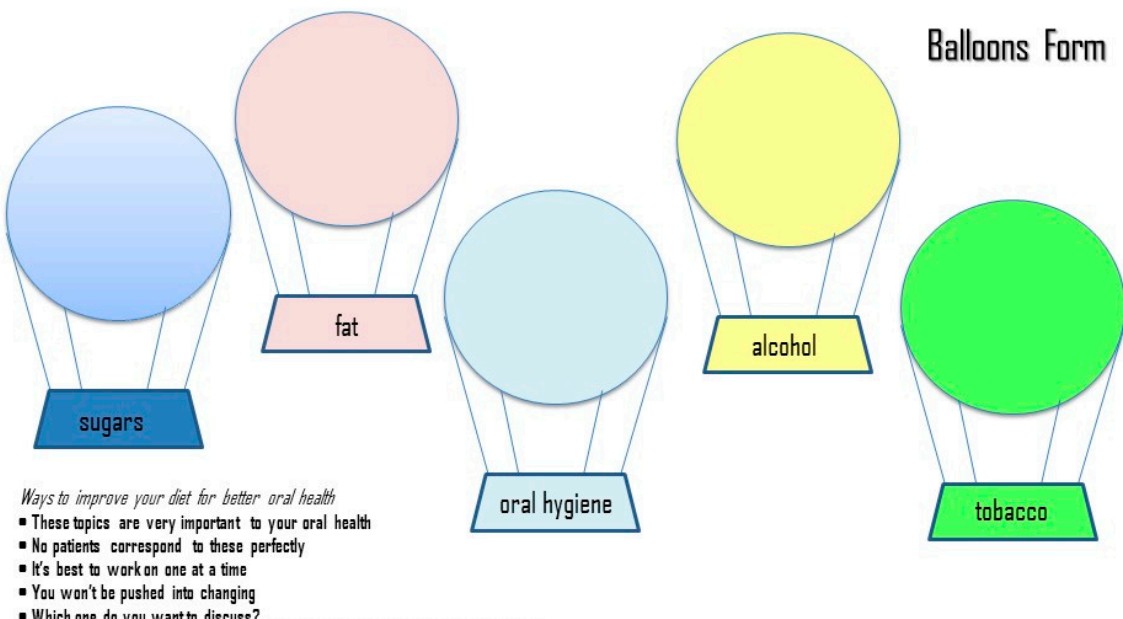

**Figure 1.** Balloons' form. Adapted for diet oral health coaching from [145,146].

Ask-Tell-Ask-Close the Loop

In this model, the coach should ensure OA understanding and recall. This must be done in a way that the patient feels respected, accepted, and asked rather than being told to change. It should be effective for strong-minded OA who do not accept their impairments due to aging.

Ask permission: The provider asks permission to give information about a topic of importance to the patient (*"Would you like to hear more about.."* or *"Is it ok if I share some information about the importance of physical activity?"* or *"I'd like to show you how to check your teeth. Would this be a good time?"* or *"There are several things I want to tell you about the new periodontal therapy. Ready?"* or *"Can we talk a bit about your (insert risky/problem/unhealthy behavior)?"* or *"I noticed that you have (insert conditions). Do you mind if we talk about how different lifestyles affect (insert condition)?"* (Diet, exercise, smoking, and alcohol use can be substituted for the word "lifestyles" so as not to provoke guilt).

*Tell:* Explanations and written material are most effective when given in response to the patient's expressed agenda and tailored to their ability to understand. Simple visual aids can be very effective. Powerful questions might be the following: *"Do you mind if we spend a few minutes talking about … ?"* or *"What do you know about … ?"* or *"Would you like to share with me your emotion on this matter?"* or *"What do you know about how your (insert a health behavior) affects your (insert health problem)?"* or *"Are you interested in learning more about … ?"* or *"What do you know about the benefits of including Med-Diet in your diet plan?"* or *"So you said you are concerned about gaining weight if you stop smoking; how much do you think the average person gains in the first year after quitting?"*

*Ask for understanding:* The dentist then should provide information, considering the following tips: (1) Address gaps in understanding; (2) use language the patient can understand and avoid jargon; (3) share information in small bits, tailored to patients' questions or concerns; (4) use graphics, charts, models leaflets, etc.; (5) monitor whether the patient is tracking nonverbally; (6) encourage family/significant other involvement; finally, (7) ask for understanding (*"What questions do you have?"* or *"Please tell me what do you now understand about diet and how you think we need to proceed to get this under control?"* or *"When you go home, what will you say to (family member, other caregiver) about what we talked*

*about today and what you plan to do?)"* In this point, one should be aware that people often have either little or incorrect information about their behaviors. Research has shown that telling people what to do does not work well [142,146]. Most individuals prefer to be given choices in making decisions to change behaviors. By presenting information in a neutral and nonjudgmental manner empowers a person to make informed decisions about quitting or changing a risky/problem/unhealthy behavior [146].

　　*Close the loop:* The physician asks the patient to restate the information as the patient understands it. The provider can then tailor the information for the patient's needs and level of understanding. It is important to mention here that, as reported elsewhere, diabetes patients recalled and comprehended only 12% of new concepts introduced during the visit. Those patients whose recall and comprehension were assessed were more likely to have hemoglobin A1c levels below the mean [147]. Finally, the Closing the Loop technique was not found to add time to the visit duration [147] and, as such, could be of service at a busy dental office or senior center.

Rating of Change Check

　　It is said that patients are more likely to succeed with a health behavior change when the change can be related to a matter that is important to them and other people they care about (grandchildren, friends in certain activities, etc.) and when they are confident that they can achieve the change. This is a point that is very important for OA who feel their competencies to decrease. Simple ratings of level of importance and confidence using a scale of 0 to 10 can give a quick indication of readiness and next steps. Ratings of less than 7 on either measure signify less likelihood of success and the need to explore concerns and barriers with the patient, or even to select a different topic for health behavior change [10,122]. In this modality, it is important to know that there should be avoidance of the use of scare tactics, lectures, or direct warnings, as some people might pretend to agree in order to not be further attacked. Effective questions on this part of the session would be: *"Why would you want to make this change?" or "How might you go about it, in order to succeed?" or "Do you think it will help you to have strong teeth when you talk or smile?" or "How about being able to talk without embarrassment of having no teeth in front of your friends/grandchildren?" or "Do you understand that by eating so much sugar you'll end up having no teeth when you will need them most? "What are the three best reasons for you to do it?" or "How important is it for you to make this change?" or "So what do you think you'll do towards . . . ".*

Goal Setting and Action Planning

　　Goal setting and action planning can be performed by filling up action planning forms, which are problem-solving forms addressing patients' barriers to achieving success with behavior change and put them in action. Once a health goal is chosen that is important and meaningful to OA, the next step is collaborative work with the dentist/coach to create an action plan, framing small steps that have high likelihood of success. In one study, action planning was found to take as little as 1 or 2 min or as long as 20 min. The average was 6.9 min [137]. Some OA may require a longer visit or additional contacts to help achieve their self-management goals. However, most of them would be best served by a short process that is revisited, improved, and modified over short periods of time due to memory loss. Questions that might be asked in this modality are: *"When will you know you have succeeded in your goal?" or "If you had a magic wand, what would you ask for first?" or "How do you imagine yourself in six months' time concerning (mention the problem)?"*

Problem-Solving Check List

　　The problem-solving check list provides a quick assessment tool for grading the steps needed in order to change a habit. OA should write down, with the help of the dental professionals, the identified problems and the proposed solutions in order to overcome barriers [122]. (1) *"On a scale of 0 to 10, with 0 being not at all confident and 10 being as confident as you can be, how confident are you that you can (describe the activities on the action plan here)?"*. Depending on the patient's answer, ask follow-up questions. *("What makes you say 6?" or "What led you to rate it as high as a 6?" or "What has helped you to*

*be confident in the past?" or "What might help you get to a 7 or 8?"or "What could I do to help you feel more confident?")* (2) Anticipate barriers and consider strategies to overcome them. (*"What might get in the way of completing your action plan?" or "Anything else?" or "What might help you to overcome... (barrier)?" or "What has helped in the past?" or "What else?" or "What or who might help you?" or "Here is what others have done..."or "How will having no teeth interfere with being with your friends?" or "What keeps you away from the possibility to talk and smile with safety with your friends/grandchildren?").*

Follow-Up Strategies

In this modality, OA will be assisted in completing a checklist action plan form and given then a copy to take home. ("This form has helped many people begin to make healthy changes by spelling out small, doable steps and anticipating problems. I see you have decided with Dr. X to work on being more active. Would you be willing to work with me to complete the form and establish goals for becoming more active?" or "I'd like to call to see how you are doing with your action plan. Would that be OK with you? When would be a convenient time?" or "What would you like to do in the next few weeks on behalf of your diet?" or you could assess how convinced the patient is for the change "On a scale of 0 to 10, with 0 being not at all confident and 10 being as confident as you can be, how convinced are you that it is important to (insert patient's goal)?" and depending on the patient's response one might say, "What makes you say 3?" or "Why 3 and not zero? or "What might lead you to rate this as a 4 or 5?" or "What would have to happen for you to rate it higher?" or "How about raising it one point higher?" (prefer small raises that do not scare the patient).)

Collaboratively set goals. Creation of an action plan and high OA confidence for making behavior changes are not, of course, enough to guarantee healthy change. Follow-up with patients, during subsequent visits and between visits, to assess progress and adjust plans as needed is an essential part of self-management support [148–150]. The establishment of healthy habits, like getting enough sleep, choosing Med-Diet food, staying in touch with family and friends to keep the spirits up, eating in company, joining a walking group or other social groups, and surrounding oneself with loved ones and happy people, should be checked on follow-up [144].

Laser Coaching in OA

In this rapid "laser effective" coaching approach [151], the following steps have to be carried out in almost a 15–20-min session: (1) Giving permission to do coaching (*"Would you like coaching on this subject*?), (2) helping make the results clear (*"What do you really want? How will you know when it has been achieved? How do you have success in mind?"*), (3) identification of the importance (*"Why is this so important to you*?"), (4) identification of the consequences, in case no action is taken (*"What is the cost, does it cost, or will it cost you if you continue on the same path?"*), (5) identification of the obstacles (psychological, emotional, natural*) ("What limits you so that you do not face the situation? What excuses or rationalizations have you used to prevent you from moving forward*?"), (6) decision and taking action as the very next step ("*What is the next step* (try to give a simple action) *that will motivate you as quickly as possible?* (Today!) (within the next 15 min)), (7) giving responsibility ("*Apart from me, who or what else can you use as a lever to ensure that you continue your commitment*?"), and (8) recognition and reward. This final step is the most important of the model since it supposes to have a direct impact in helpless and frustrated, due to mind impairments, OA.

Ask Me Three Questions Model

This model is also an elegant quick oral health model approach since it is based mainly in only three questions that the patient should ask: (1) What is my main problem? (2) What do I need to do? (3) Why is it important for me to do this? So basically, it can be effective in a 15–20-min coaching session and can be applied right before or after the dental therapy [122,152]. It is comprised of the following questions, in the mentioned order: *"What is most important for you to accomplish during your visit today*?" (agenda); *"What do you think is the problem here*?" (knowledge, beliefs); *"What ideas do you have about*

*what is contributing to your problem*?" (knowledge, beliefs); *"What ideas do you have about treatment or things you can do to manage your condition?"* (knowledge, preferences); *"How important do you think it is to do . . . ?"* (X treatment or self-management task) or *" . . . to manage or treat your condition?"* (ideas, values, preferences); *"What would you like to know about your condition?"* (knowledge, Preferences); *"What concerns you the most about your condition*?" (feelings); "How do you feel about trying to . . . ?" (feelings); *"What specifically would you like to work on to manage your condition*?" (goals); *"What is that you want for yourself in six months' time?"* (goals); *"What would help you to manage your condition*?" (needs, preferences); *"Who do you think will help achieve this?"* (needs, preferences); *"How confident are you that you could do* (X treatment or self-management task)?" (feelings); *"From 1 to 10, how much you believe you can achieve* (X treatment or self-management task)?" (feelings); *"What might get in the way or keep you from being successful*?" (barriers); *"How do you think you can surpass this?"* (knowledge, beliefs).

All proposed models and their effects and characteristics for the OA are seen synoptically in Table 3.

**Table 3.** Diet and oral health coaching models for OA.

| Model | Method | Type of OA | Results |
|---|---|---|---|
| OARS | Open-ended questions, affirmations, reflections, summary | OA who like talking and communicating with others | Discovery of goals, clarification of wishes, acknowledgement of contradictions, strengthening patient' own motivation |
| Dental PAM | 13-questions Questionnaire | Ambivalent, fearful, uncertain, reluctant to change, untrusting OA | Evaluation of knowledge, skills and confidence Stratification of patients according to activation level |
| Tell-Show-Do | 3 simple and quick steps: share information, show how to do it, let patient do it | OA with physical impairments or with short memory loss | Quick evaluation of perceived information, achieving results through often repetition and exercise |
| Balloons Diagram | Balloons form | OA with hearing or other physical difficulties, optical way of learning | Sudden realization, visualization, metaphorical release of problematic situations and habits |
| Ask-Tell-Ask-Close the Loop | Ask for permission, give information through written materials, brochures, etc., ask for understanding and rephrase goals | OA with sensitivities, depression, negative feelings, isolated, strong-minded, unwilling to accept age impairments | Specification of goals, feeling trusted and accepted |
| Rating of Change Check | Change Check List | OA who likes numbers and numerology | Determination for achieving small steps, summarization of change |
| Goal setting and Action Planning | Action Planning Form | OA who still can write, with good vision but memory loss, those who like order and organization | Stratification and empowerment of goals, strengthening of motivation |
| Problem Solving | Problem-Solving Check List | Impatient, stressed, economic dependent OA | Lower guilty behavior and stress |
| Follow-ups | Follow-up Check List | Lonely OA with memory loss or fear of incompetence | Self-acceptance and lower stress |
| Laser Coaching | Short, compact communication based on reward | For OA who need recompense and like prizes, bonus, presents and gifts | Giving responsibility, feelings of self-realization and value |
| Ask-Me-3-Questions | The patient makes the questions and the answers | For reluctant OA | Accountability |

## 7. Discussion

So far, it has been shown that individually tailored oral health education program is better than traditional education [135,153]. Thus, the after-sales' service is very important in the situation of OA. All over the relevant literature discussed already, the dental and medical professionals should organize follow-up support to help OA sustain healthy behaviors between visits. They should be persistent in short breakthroughs and often follow-ups due to the memory issues of OA. They should also extend care into the community by linking elders to community programs. They should further build a team of people trained to make coaching interventions and assign responsibility for self-management tasks to all team members, extending the work out from the dentist. Finally, they should use daily team huddles to review the schedule of patient charts, anticipate care needs, and enhance the flow of care in an aging population.

Sustaining healthy behaviors for a lifetime requires courage and tenacity, most often involving small, incremental changes that build over time into bigger successes. Even the best plans of action require adjustment from time to time in order to work effectively. For these reasons, making regular contact with OA after a visit or change in diet protocol or dental treatment is central to sustaining positive change. Studies in depression document the need to follow up with patients to assist them in succeeding with their action plan. Helpful as it seems might be the connection of OA with sources of support in the community such as recreation or senior centers, support groups, and voluntary community organizations. Finally, quite appealing might be to locate or develop a peer program in the dental clinic or community involving them actively with other people.

Possession of preventive knowledge and skills alone will not ensure the OA's attainment of the goal of preventive counseling, that is, maintenance of optimal diet and oral health status. The dental professional and patient must establish a therapeutic alliance, whereby each is committed to performing the activities necessary to achieve this goal. OA must be convinced that ultimately only they can help themselves by adhering to the recommended preventive measures. It might thus be helpful if the provider frames his or her oral health messages in terms of the senior patient's overall health, as this may lend to more trust, credibility, and urgency for the patient to take such messages seriously and, finally, act [98,100,130].

Professionals should work to dispel the misconception that oral disease is an inevitable consequence of aging, and that, consequently, the attempt to prevent oral disease is a futile effort. Park and Chang [24], mentioned that change comes not only by the capacity of the participants to engage in behavior change but also on the performance of the individual health coaches. According to the spirit of MI, the therapeutic relationship is more like a partnership or companionship than expert/recipient roles [140]. It is, therefore, essential that health coaches are supported in their role. It is recommended also that adequate training budgets and adequate reimbursement of health care providers for their time and commitment will help with the sustained recruitment of program participants, the effective running of these types of programs, and, ultimately, the outcomes [24].

Thus, the oral health services, dental schools, and medical faculties should be organized and developed to secure adequate early detection and prevention as well as treatment of oral health problems for all OA, whether living at home or in hospitals and health care facilities. The achievement of such a service goes beyond what a dentist can do alone. It requires the involvement of other health professionals and health care workers. This presents a realistic goal that could assure good quality of life and a reduction in the dental expenses for the elderly patients.

It is suggested that dentists should implement health coaching programs as a package in their services, containing coaching on diet and oral health prevention, goal setting, attainment, and adherence promotion. In addition, respecting each participant's autonomy and resisting the urge to push against any resistance put up by them, dentists might have a better chance to reach positive outcomes [142,154]. In the study of Park and Chang [61], participants reported a high level of goal achievement. The results are consistent with previous studies for OA with multi-morbidity where it was reported that health

coaching intervention enhanced residents' participation in intervention programs, resulting in a significant increase in their self-efficacy and self-management behaviors [155].

The health provider thus becomes a colleague, offering guidance and support instead of solely telling patients what to do to manage their oral health. In the context of a collaborative relationship with shared decision making, dental professionals can provide the elements of self-management support, including self-monitoring and problem solving, goal setting, action planning, and rewarding. To reinforce this outcome, it is interesting to know that when patients receive collaborative self-management support, they have fewer hospitalizations, improved quality of life, and improved clinical outcomes in several ambulatory-sensitive conditions [81,82]. Further, it has been shown that a short form that elicits patient concerns or needs, either mailed in advance of the visit or completed in the waiting room, can be sufficient [156,157]. Patients often leave the office visit without understanding or remembering important care instructions and medication information [158], which may lead to worse outcomes such as higher hospitalization rates [159]. Twenty percent of patients read at a fifth-grade level or below, for which written health care information is not often tailored. Physicians cannot expect that patients will spontaneously reveal their lack of understanding. Also, physicians may not provide basic information that patients need. In one study, physicians explained the adverse effects of medications or instructions about one-third of the time [160]. Despite these data, by using simple methods of coaching, like the ones mentioned here, a senior nutrition and oral health coach can help improve communication and patient understanding towards healthier nutritional habits that correspond to the OA-specific needs.

So, it seems that for the diet-changing behavior and oral health prevention scope, the team members and care givers should ensure the practical and psychological part of a good meal in order for the elderly to keep enjoying food despite any physical impairment or "being on your own", highlighted in many studies [161–163]. It is promising in this way, the fact that current older patients are better educated, more politically aware, and have more remaining teeth than in previous generations [164]. However, the older population is not homogenous. OA who have lower incomes have poorer oral health and more limited access to services [165], even more to senior coaching sessions, a fact that should sensitize the political leadership nationwide.

In the economic recession period that will follow the COVID-19 crisis, care givers, nurses, dentists, and other medical professionals should find their original motive in doing what elderly care needs despite the practical and economic difficulties and should be urged through coaching to estimate their values in taking care of the elderly. People who love others and have a good level of emotional intelligence should be better candidates for elderly units and dental offices for seniors [7,96,166].

In conclusion, dental and other medical professionals should reevaluate their role as health coaches in order to improve dietary habits and nutritional intake of the OA. By reminding themselves that dentistry is a helping profession, they will see more value in "oral health coaching" as a desired and supportive means to an end. In fact, they are helping people to make decisions that can add to the quality of their lives. By altering their thinking and approach slightly they can easily shift the focus from "us" and our procedures to "the patient" and the quality-of-life impact their services can have on their lives. This shift in thinking will enable them to communicate with their senior patients in a more mentorship-based, collaborative, and inspiring way.

Modern dentistry is bright and filled with opportunities when someone chooses to expand his/her clinical excellence while concurrently taking the time to grow as an "oral health coach". "Oral health coaching" is the emerging yardstick that will differentiate professionals, especially dentists, to become more effective as clinicians while feeling more trusted and valued in the eyes of their patients.

## 8. Conclusions

In this nonsystematic review the process of senior coaching on diet issues for better oral health were discussed for the independent elderly *or older adults (OA),* referring to individuals of age 65 or older. It can finally be concluded that:

(1)　There are certain mental and physical issues resisting change in habits and behavior of OA.

(2)　OA are more likely to benefit from a series of quick health education sessions followed by tailored feedback that is based on the absence of criticism, patience, empathy, and total acceptance by the dentist/professional coach.

(3)　Overcoming persistent noncompliance of OA through specific educational training can make health-behavior change one of the most rewarding and the most challenging responsibilities for dental health professionals.

(4)　Coaching models based on filling out forms or lists of goals, tasks, recruiting small steps, and rewarding are suggested as being more effective in OA due to their mental and physical issues.

(5)　Health professionals should reevaluate their role as health coaches in order to improve dietary habits and nutritional intake of the OA.

(6)　"Oral health coaching" will enable professionals to communicate with their senior patients in a more mentorship-based, collaborative, and inspiring way.

**Author Contributions:** Authors declare their equal contribution in the work reported. All authors have read and agreed to the published version of the manuscript.

**Funding:** This research received no external funding.

**Conflicts of Interest:** The authors declare no conflict of interest.

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
