# Peer review of "Diet and Oral Health Coaching Methods and Models for the Independent Elderly"

_applsci, doi:10.3390/app10114021_

Round 1

Reviewer 1 Report

The paper reviews diet and oral health coaching in the independent elderly. It is a well-written and very interesting manuscript, covering many aspects on the specific topic.
I suggest it is accepted for publication.

Author Response

Thank you very much

Reviewer 2 Report

The review was written very well by the author. Correct some typo errors.

Author Response

Typo errors have been corrected

Reviewer 3 Report

In the manuscript "Diet and Oral health Coaching in the independent elderly" authors present suggestions and models for relevant senior coaching interventions.

Therefore health-related behavior based on diet and hygiene are the important determinants of oral health in older adults and require extensive research.

This study is very interesting and after minor changes, I will recommend being published. I feel there are a number of methodological considerations and presentation issues that need to be addressed.

The most important proposal is to change the title. When I started reading the manuscript, I thought it would include dietary and hygiene guidelines that should be considered when educating an elderly patient. At the same time, to my great surprise and pleasure, the article is an excellent review of the methods that can be used to successfully conduct this education. The methods are described in detail with practical tips. This comprehensive approach suits me very much and constitutes a great value of work. But it should be presented in the title!

However, special attention must be paid to plagiarism issues when using descriptions of methods and models.

Another element that I suggest to add to the work is a graphic or tabular presentation of some information. A long text is difficult to read and systematize the information. Also tables, lists, diagrams are more often analyzed and presented by scientists, and this will increase not only the attractiveness of the article, but also its citation.

I propose to:

  • introduce a table presenting Physical and mental issues of OA that resist change of attitude.
  • as a diagram, I suggest presenting the text contained in lines 120-125.
  • and also to develop a large comprehensive model presenting the entire algorithm of conduct, input data (needs and limitations of seniors), available methods, models and methods of their selection, techniques, issues related to motivation and ultimately results.

English is really good, but there are also some stylistic problems. The sentences presented below are too long or unclear:

  • 37-42 In response to these challenges, on the other part of the Atlantic, the European College of Gerodontology (ECG) and the European Geriatric Medicine Society (EUGMS) have created a common task and Finish Group which reported that the development of a workforce of dentists with knowledge about and skills for working with ΟΑ would be enhanced by interdisciplinary and interprofessional education [4], a philosophy also suggested by others in the past [5-7].
  • 44 working yet aging in a functional way
  • 168-171 Many of them, usually quite independent during their lifespan, may be embarrassed about the need for help after some years of age or may lack resources to make changes or think they cannot afford some of the desired interventions, and may fear failure and the associated perception that they are incompetent.

The bibliography is very extensive, the formatting is correct. It shows a good knowledge of this subject by the authors.

Author Response

In the manuscript "Diet and Oral health Coaching in the independent elderly" authors present suggestions and models for relevant senior coaching interventions.

 Therefore health-related behavior based on diet and hygiene are the important determinants of oral health in older adults and require extensive research.

This study is very interesting and after minor changes, I will recommend being published. I feel there are a number of methodological considerations and presentation issues that need to be addressed.

The most important proposal is to change the title. When I started reading the manuscript, I thought it would include dietary and hygiene guidelines that should be considered when educating an elderly patient. At the same time, to my great surprise and pleasure, the article is an excellent review of the methods that can be used to successfully conduct this education. The methods are described in detail with practical tips. This comprehensive approach suits me very much and constitutes a great value of work. But it should be presented in the title! (there has been changes in the title)

However, special attention must be paid to plagiarism issues when using descriptions of methods and models.(changes have been made)

Another element that I suggest to add to the work is a graphic or tabular presentation of some information. A long text is difficult to read and systematize the information. Also tables, lists, diagrams are more often analyzed and presented by scientists, and this will increase not only the attractiveness of the article, but also its citation.

I propose to:

  • introduce a table presenting Physical and mental issues of OA that resist change of attitude.(inserted table 1, page 3)
  • as a diagram, I suggest presenting the text contained in lines 120-125.(inserted table 2)
  • and also to develop a large comprehensive model presenting the entire algorithm of conduct, input data (needs and limitations of seniors), available methods, models and methods of their selection, techniques, issues related to motivation and ultimately results.( Table 3 was inserted)

English is really good, but there are also some stylistic problems. The sentences presented below are too long or unclear:

  • 37-42 In response to these challenges, on the other part of the Atlantic, the European College of Gerodontology (ECG) and the European Geriatric Medicine Society (EUGMS) have created a common task and Finish Group which reported that the development of a workforce of dentists with knowledge about and skills for working with ΟΑ would be enhanced by interdisciplinary and interprofessional education [4], a philosophy also suggested by others in the past [5-7]. (splitted in smaller phrases)
  • 44 working yet aging in a functional way (corrected)
  • 168-171 Many of them, usually quite independent during their lifespan, may be embarrassed about the need for help after some years of age or may lack resources to make changes or think they cannot afford some of the desired interventions, and may fear failure and the associated perception that they are incompetent. (corrected)

The bibliography is very extensive, the formatting is correct. It shows a good knowledge of this subject by the authors.

Reviewer 4 Report

This review explores a field that only few articles have been studied and the authors are doing a great job. They explained the difficulties of and the necessity for diet and oral health coaching especially for the group of older adults. They also provided and explained practical approaches to achieve the goals. This manuscript is in a good form for publishing, and just needs a minor English check. 

Author Response

Minor English check has been carried out